# Responding to the Heat and Planning for the Future: An Interview-Based Inquiry of People with Schizophrenia Who Experienced the 2021 Heat Dome in Canada

**DOI:** 10.3390/ijerph21081108

**Published:** 2024-08-21

**Authors:** Liv Yoon, Emily J. Tetzlaff, Carson Wong, Tiffany Chiu, Lucy Hiscox, Samantha Mew, Dominique Choquette, Glen P. Kenny, Christian G. Schütz

**Affiliations:** 1School of Kinesiology, Faculty of Education, University of British Columbia, Vancouver, BC V6T 1Z1, Canada; carsonjw@student.ubc.ca (C.W.); lucyhx@student.ubc.ca (L.H.); sammew@student.ubc.ca (S.M.); dmchoq03@student.ubc.ca (D.C.); 2Human and Environmental Physiology Research Unit, School of Human Kinetics, Faculty of Health, University of Ottawa, Ottawa, ON K1N 6N5, Canada; etetz085@uottawa.ca (E.J.T.); gkenny@uottawa.ca (G.P.K.); 3School of Population and Public Health, Faculty of Medicine, University of British Columbia, Vancouver, BC V6T 1Z3, Canada; tchiu32@student.ubc.ca; 4Clinical Epidemiology Program, Ottawa Hospital Research Institute, Ottawa, ON K1H 8L6, Canada; 5Department of Psychiatry, Faculty of Medicine, University of British Columbia, Vancouver, BC V6T 2A1, Canada; christian.schutz@ubc.ca; 6Mental Health and Substance Use Services, Provincial Health Services Authority, Vancouver, BC V5C 6E3, Canada

**Keywords:** climate change, air conditioning, extreme heat, heat wave, ill-housed persons, mental disorders, mental illness, risk reduction behaviour

## Abstract

People with schizophrenia have died at disproportionately higher rates during recent extreme heat events (EHEs) in Canada, including the deadly 2021 Heat Dome in British Columbia (B.C.). However, to date, little research has qualitatively focused on how people with schizophrenia experience and respond to EHEs. This study aimed to (i) explore how people with schizophrenia experienced and were impacted by the 2021 Heat Dome physically, cognitively, and emotionally and (ii) understand their level of awareness and health-protective actions taken in response to the EHE. Between October 2023 and February 2024, interviews were conducted with 35 people with schizophrenia who experienced the 2021 Heat Dome in a community setting within B.C., Canada. The semi-structured interviews were guided by pre-defined questions to explore the participant’s background, living situation, social network, awareness and access to heat-mitigation measures. The transcripts were analyzed using a descriptive form of thematic analysis. Participants shared critical insights on how the EHE impacted them, including descriptions of mild to severe physical manifestations of heat stress (e.g., fainting, heat rashes), the triggering of schizophrenia-related symptoms (e.g., paranoia, hallucinations), and the detrimental effects on their energy levels and emotional stability, which further caused disruptions to their everyday life. Participants also illustrated gaps in knowledge and challenges experienced with accessing information, which hindered their ability to manage the heat exposure effectively and, for some, resulted in no actions (or counter-intuitive actions) being taken to mitigate the heat. These findings demonstrate the complex ways that individuals with schizophrenia experienced and responded to the 2021 Heat Dome and revealed various situational and contextual factors that further compounded the challenge of heat mitigation. These findings can support the development of tailored individual and community-level heat response and communication initiatives and strategies for people with schizophrenia.

## 1. Introduction

People with schizophrenia and schizoaffective disorder (referred hereon as ‘schizophrenia’) have died at disproportionately higher rates during recent extreme heat events (EHEs) in Canada—defined as “two consecutive days of weather that meets or exceeds the criteria set for the region’s daytime highs and nighttime lows” [1]. For example, of the 619 deaths in the registry for the 2021 Heat Dome in British Columbia (B.C.), Canada, 16% (*n* = 97) had been diagnosed with schizophrenia [2,3]. Similarly, in the 2018 EHE in Montreal, people with schizophrenia accounted for 26% (*n* = 17) of heat-related deaths [4]. Given that less than 1% of Canadians are diagnosed with schizophrenia [5], these numbers pose a critical concern. This impact has also been seen globally. For example, Sung et al. [6] conducted a 12-year study to evaluate the relationship between the mean daily range of ambient temperature and hospital admissions for schizophrenia in a national cohort of psychiatric inpatients in Taiwan and found a joint effect of temperature associated with elevated morbidity risk in extremely hot conditions. Moreover, the relationship between air temperature and schizophrenia hospital admissions, particularly in arid urban climates, has been quantitatively affirmed [7]. A significant portion of hospitalizations has been attributed to high-heat conditions, underscoring public health and economic burdens [8].

Schizophrenia is a chronic and severe mental disorder that affects how a person thinks, feels, and behaves. People suffering from schizoaffective disorder, in contrast to those suffering from schizophrenia, are additionally affected by manic or depressive episodes. These disorders are generally diagnosed by psychiatrists using diagnostic criteria from the Diagnostic and Statistical Manual of Mental Disorders (DSM). Core diagnostic symptoms include hallucinations, delusional or disorganized thinking. The symptoms must be which are persistent and impact the individual’s functioning. There appears to be a diagnostic continuum of symptoms consistent with schizophrenia: without any affective disorder symptoms (e.g., mood disorders), with some affective symptoms, with psychotic symptoms consistent with schizophrenia with affective episodes (schizoaffective disorder), and with psychotic symptoms consistent with schizophrenia without affective episodes (schizophrenia). This study included individuals diagnosed with either schizophrenia or schizoaffective disorder. For simplicity, we use the term “schizophrenia” to refer to both conditions.

Previous studies have investigated the relationship between schizophrenia and heat and found that people with schizophrenia experience impairments in their ability to regulate body temperature [9,10,11,12,13,14,15,16,17]. These impairments have been largely attributed to the medications typically prescribed to manage schizophrenia [18]. These medications have been known to affect metabolic heat production and blunt blood vessel and sweat gland activity. Consequently, individuals on these medications can experience periodic elevations in core body temperature above what is expected, further narrowing the threshold for heat stress [19,20], and potentially leading to an increased risk of hospitalization during extreme heat conditions. Similarly to other heat-vulnerable populations (e.g., older adults, individuals with type II diabetes), people with schizophrenia also have a reduced ability to recognize overheating and initiate protective actions [16,17,21]. Additionally, these individuals often experience social vulnerability due to a higher prevalence of factors such as homelessness and social isolation, which reduce protective buffers, such as access to amenities and having someone check- in on them [22,23,24,25]. Together, these studies highlight the compounding factors that heighten the risk of adverse health outcomes among people with schizophrenia during EHEs.

While some studies have qualitatively focused on the lived experience of people with schizophrenia, few have assessed the impacts of extreme heat on this population [26]. Moreover, to the best of our knowledge, no interview-based studies have been conducted to evaluate how people with schizophrenia experience and cope with extreme heat. This is an important gap given that comparable qualitative interview-based studies that have explored gaps in first-hand knowledge from other populations who have survived EHEs (e.g., migrant groups [27], socioeconomically marginalized [28], elderly citizens [26,29,30], people with disabilities [26], people with substance use challenges [26], and people who are insecurely housed [31,32]) have revealed critical insights into how different vulnerable people understand, experience and cope uniquely during EHEs. Moreover, studies have also revealed how those considered most vulnerable often do not identify the vulnerability within themselves [29]. These findings highlight that heat vulnerability is not solely defined by the physiological changes associated with heat exposure, but also by the social and personal factors that can affect how a person experiences and copes.

With the increasing frequency and intensity of EHEs projected to continue in Canada, extreme heat poses a critical public health challenge. Therefore, to help address these important knowledge gaps, we sought to qualitatively assess how people with schizophrenia experience and respond to EHEs. Specifically, we performed an interview-based inquiry with those directly impacted by the 2021 Heat Dome to (i) explore how people with schizophrenia experienced and were impacted by the 2021 Heat Dome physically, cognitively and emotionally and (ii) understand their level of awareness and health-protective actions taken in response to the EHE. Through their accounts, we now identified opportunities to help shape our responses, policies, and actions to support and protect this vulnerable population during future EHEs.

### Conceptual Framework

In this study we adopt the biopsychosocial model [32,33], a comprehensive framework for understanding health and illness by integrating biological, psychological, and social factors. Unlike traditional models that focus predominantly on biological aspects of health, the biopsychosocial approach recognizes that human health is influenced by a complex interplay of genetic predispositions, mental and emotional states, and social environments. This model posits that biological factors, such as genetics and physiology, interact with psychological processes, including thoughts, emotions, and behaviour, while being shaped by social contexts, such as family dynamics, socioeconomic status, and cultural influences [33]. By addressing these multiple dimensions, the biopsychosocial model offers a more holistic view of health and illness, facilitating a deeper understanding of how various elements contribute to overall well-being and the management of chronic conditions.

## 2. Materials and Methods

### 2.1. Ethics

This study received ethical approval from the University of British Columbia (UBC) Research Ethics Board (BREB H23-01595) and the Health Canada Research Ethics Board (REB 2023-008H). Additionally, the Vancouver Coastal Health Research Institute (VCHRI) (V23-01595-2023) and the B.C. Mental Health Substance Use Services (BCMHSUS 2023-09-26) granted operational approval.

We ensured that all participants provided informed, free, and voluntary consent before participating in the study, whether in-person or virtually. Given the potential vulnerability of individuals with schizophrenia, we implemented specific measures tailored to both data collection methods to ensure ethical practices were adhered to effectively.

For in-person interviews, we provided consent forms in accessible language and offered verbal explanations of the study’s purpose, procedures, and potential risks. We made sure that participants had sufficient time to ask questions and receive answers before consenting. For those with difficulty comprehending the consent process, clinicians or health care staff were present to assist and ensure understanding.

Virtual interviews required additional steps to address the unique challenges of remote communication. We used secure video conferencing platforms and took extra time to confirm that participants fully understood the consent form, which was sent electronically prior to the interview. We provided verbal explanations during the virtual session and encouraged participants to ask questions. We also ensured that virtual consent procedures were documented and that participants had ample opportunity to express any concerns.

To mitigate potential risks, such as psychological distress from discussing sensitive topics, we implemented several precautionary measures for both (in-person/virtual) data collection methods. Participants were informed about potential risks and benefits before the interviews, and we emphasized their right to withdraw from the study at any time without consequence. We also provided information on support services should participants experience any distress.

Overall, our approach was designed to respect participants’ autonomy and protect their well-being, adapting our procedures to the specifics of in-person and virtual interactions to ensure that ethical standards were consistently met.

### 2.2. Participant Selection and Recruitment

Participants (>18 years of age), with a clinical diagnosis of schizophrenia and who experienced the 2021 Heat Dome, were recruited from in-patient and community settings within B.C., Canada. People residing within care facilities at the time of recruitment (e.g., Red Fish Healing Centre, Vancouver Coastal Health (VCH), UBC Hospital) were recruited in-person with the support of site staff and clinicians. Based on the setting and site-specific requirements, this occurred via research staff directly approaching participants during programming breaks or receiving names of qualifying and interested patients from site staff. We approached 31 people for in-person recruitment, and 28 agreed to participate (*n* = 19 inpatients at Red Fish Healing Centre [part of BCMHSUS]; and *n* = 9 inpatients who currently live in relevant VCH clinics). Additionally, 30 people who had previously resided as in-patients or accessed these services from community settings were invited to participate through letters mailed to their current residence; no responses were received. Digital posters were also circulated on social media and through email lists from associated community organizations (e.g., B.C. Schizophrenia Society, VCHRI, BCMHSUS); *n* = 7 people were recruited through this method. In total, 35 people took part in this study.

Eligibility criteria were applied consistently across recruitment sites. For in-person interviews, site staff assisted with recruiting participants and confirmed their eligibility through checking patient charts, listing their age and clinical diagnosis. Prior to finalizing recruitment, research staff confirmed the patient had experienced, and could recall, the 2021 Heat Dome. However, it is important to address that recall challenges are a prominent symptom of schizophrenia, and thus retrieval of memories more than two years after the EHE may be vague, partial, or possibly inaccurate. As such, our study design more broadly allowed for insights into the experiences of people with schizophrenia under general heat stress, such as the 2021 Heat Dome, to be captured.

For virtual interviews, individuals who volunteered to participate were screened for eligibility by the research staff. Participants were asked if they were over 18 years old, had a clinical diagnosis of schizophrenia, and were residing in B.C. during the 2021 Heat Dome (as a proxy for experiencing the 2021 Heat Dome).

### 2.3. Interviews

The semi-structured interviews were conducted between 13 October 2023 and 22 February 2024 (L.Y., C.W., S.M., D.C., L.H., T.C., E.J.T.). All members of the research team underwent extensive ethical and interview training with VCHRI, BCMHSUS and attending clinicians on appropriate, safe, and sensitive ways to engage with people with schizophrenia. The interviews were conducted with a single study participant and one or two research team members. Following each interview, research members wrote reflections to note any cues that could not be captured by the audio recorder, such as body language and tone of voice. The semi-structured interview guide included 26 primary question prompts, categorized into six topics: participant background, recollection of the 2021 Heat Dome, medical profile, living situation, built environment and social network, mitigation measures to keep cool during the heat, and needs for future heat-protection. The lead author drafted an initial set of questions aimed at exploring the core experiences of participants, structured to cover the main themes identified in the literature about heat vulnerability thus far. The questions were reviewed by experts in the field, as well as psychiatrist team members, which helped refine the questions for clarity, relevance, and accessibility. We piloted the guide within the team in a mock interview led by a psychiatrist and made necessary adjustments to the wording, sequence and number of questions (Appendix A).

The interviews were completed in-person at a care facility(*n* = 28), virtually over the UBC campus-wide Zoom license (*n* = 5), or by phone (*n* = 2). The same semi-structured interview guide and interviewing procedures were followed across interview settings to ensure consistency in data collection and consequent data analysis. For in-person interviews, private spaces were secured in the associated care facility (e.g., research room) based on the participant’s preference and availability. For virtual interviews, participants could join from any setting in which they were comfortable; however, additional security precautions were taken to enhance privacy (e.g., password-protected meeting links strictly shared with only the research participant, the researcher maintaining control of all Zoom functions, and locking the meeting once it had begun).

Participants were interviewed once, and the interviews lasted 36.5 min on average (range: 15–120 min). Interviews conducted virtually by Zoom or phone lasted longer on average (58.5 min) in comparison with interviews conducted at a care facility in-person (26.5 min). The participants who were capable of partaking in virtual interviews were more mentally stable and independently functioning than the participants interviewed in care facilities. Additionally, as the virtual interviews were scheduled in advance, participants had the opportunity to think about their experiences in EHEs, whereas participants in care facilities were not provided with notice. This systematic difference in virtual and in-person interviews may have impacted the quality of data collection since virtual participants tended to provide greater breadth of perspective and thoughtful responses than in-person participants.

The interviews were either audio-recorded and transcribed verbatim using the software Otter.ai (version 2023), a speech-to-text transcription application using artificial intelligence and machine learning (Otter.ai, Inc., Mountain View, USA, 2016) (*n* = 32), or were recorded via manual note taking (*n* = 3). Manual notetaking entailed transcribing excerpts from the interview verbatim, when participants did not wish to be audio-recorded or when it was too difficult to understand them. These notes were made to the best of the research team’s abilities. While this strategy may have yielded less rich data than audio-recorded interviews, we felt that these interviews were important to keep in the data set as this process enables the most diverse range of perspectives to be incorporated into the analysis. The research team reviewed each audio file and transcript for consistency and accuracy (L.Y., C.W., S.M., D.C., L.H., E.J.T., T.C.). Two participants opted in for the member-checking process, for which the first author (L.Y.) presented them with the full transcript for their review and approval. All identifying information was anonymized/pseudonymized as appropriate.

### 2.4. Data Analysis

The data were analyzed using a descriptive form of thematic analysis, an effective method to identify repeated patterns of meaning across a dataset [34]. The analysis was theoretically underpinned by constructivism, with an inductive orientation that allowed for the themes identified to be strongly linked to the data (i.e., the participants’ experiences). It also took a semantic approach, as the themes developed were based on the explicit or surface meanings of the data [34]. To conduct the thematic analysis, the six steps outlined by Braun and Clarke [35] were followed using Dedoose, a qualitative content analysis software (Version 9.0.17, SocioCultural Research Consultants, LLC, Los Angeles, CA, USA). Familiarization with the data was initially performed during the transcript verification process and subsequently using repeated active reading of the transcripts (i.e., reading while searching for meanings and patterns), and initial codes were generated [36] (L.Y., C.W., S.M., D.C., L.H., T.C., E.J.T.).

Themes were identified by organizing the specified codes and collating relevant data references [37]. Coded data extracts were assigned to as many different ‘themes’ as they fit into. The themes were subsequently refined by rereading all the collated extracts and considering whether they formed a coherent pattern (L.Y. and E.J.T.). Each theme was then assessed to determine if and whether it accurately reflected the meanings evident in the data set as a whole [38]. Reflexive memos and written reflections were created alongside the coding process to help make sense of participant experiences [37,38,39]. The resulting coding structure included 4 primary themes, 11 concepts and 51 sub-concepts. The coding structure is visualized in Table 1. The full codebook, which lists each theme, concept and sub-group, also provides our definition for each, identifies the code type and code origin, and an excerpt from the interview transcripts to illustrate the theme/concept is available in Appendix A.

## 3. Results

### 3.1. General Participant Characteristics

Overall, we interviewed 35 individuals with schizophrenia (female: 12, male: 23). The average age of participants was 41 ± 17 years (median: 36 years, range: 21–80 years), and most indicated their marital status as single (inclusive of those who indicated they were divorced or widowed) (*n* = 26). Most participants were primarily securely housed during the 2021 Heat Dome (*n* = 28); however, a few (*n* = 6) participants were experiencing homelessness during the EHE (note: one participant did not report their housing status). All participants took prescribed medication(s) during the 2021 Heat Dome, and all but two used additional non-prescribed substances during that period (e.g., cocaine, heroin) (Appendix A). The legend following each participant ID indicates their age and sex at time of interview, as well as their housing status, and use of prescribed and/or non-prescribed medications and substances during the 2021 Heat Dome).

### 3.2. Theme I: Personal Reflections on the Impact of Extreme Heat

Through the interviews, our participants shared critical insight into their experiences living with schizophrenia during the 2021 Heat Dome in B.C. Firstly, the participants shed light on the nuanced ways in which schizophrenia may impact their perception of heat, affecting their ability to manage and interpret sensations of heat, their condition, and existing symptoms. Secondly, the participants provided specific examples of cooling strategies they employed to help mitigate the impacts of the extreme heat. The cooling strategies mentioned included approaches tailored for indoor and outdoor living environments at the individual- and building-levels. Thirdly, the participants proposed various heat-health-protective strategies, methods, and programs that could help support future heat protection. The following sections present these impacts and strategies with illustrative quotes from the participants’ voices. Each quote is also tagged with additional details related to the participants’ age, sex, housing status during the 2021 Heat Dome, prescribed medication and non-prescribed substance use.

#### 3.2.1. Physical Impacts of Heat

Within the interviews, it was common for participants to describe physical manifestations of the heat, including difficulty breathing, excessive perspiring, light-headedness, fainting, head rushes, dizziness, vomiting, fatigue, tiredness, feeling drained, heat rashes, and a general sense of weakness. For some participants, these signs and symptoms were described to be mild, as expressed in the following quote:


*“Kind of maybe even a bit disoriented a little bit to some degree, like not heatstroke, but like out of sorts a little bit from it.”*
(A12: 42; Male; Housed; Non-Prescribed Substance Use: Cough Syrup, Vapes, Dill, Cocaine, Acid)

However, for other participants, the experience was much more severe. For example, two participants shared that:


*“At one point in the heat dome, like I, I thought I was gonna die. Um, like, I went and had to go, but I couldn’t sleep in the living room. And I was on my home hospital bed, and it was just so hot. And it just, it felt like my soul was just gonna fall out of my body. I don’t know how else to say it.”*
(C02: age not specified; Female; Housed; Non-Prescribed Substance Use: N/A)


*“You ever been outside where it’s so hot and you feel dizzy? You just don’t feel like right? It’s not even like, any problem or anything like, you, know, it’s not your fault. It’s the outside’s fault, and you can’t stop being dizzy, and you just don’t feel like you’re even in reality anymore. That’s kind of what it was like.”*
(A10: 31; Male; Housed; Non-Prescribed Substance Use: Cannabis, Alcohol, Cocaine)

The impacts were also described as limiting physical mobility for some. For example, participants recollected heat impacting their capacity to go for walks or move:


*“I remember the air conditioning couldn’t keep up with the heat. And I remember we decided to go for a walk when it was like 41 degrees outside. And I remember being able to not really breathe very easily. And I felt really hot and very exhausted.”*
(A12: 42; Male; Housed; Non-Prescribed Substance Use: Cough Syrup, Vapes, Dill, Cocaine, Acid)


*“It was just hot, so sweaty and all my clothes were stuck to me, so I couldn’t really move around freely.”*
(A06: 25; Male; Housed; Non-Prescribed Substance Use: Cocaine, Cannabis, Alcohol;)

For some participants, their descriptions of the physical impact of heat exposure were compounded by their psychological experience. For example, one participant shared that:


*“Oh, yeah. I sweat more…it’s like, at the beginning of, a slight sensation of stress in the body because of sweating more and feeling that moisture on your skin that’s really hot feeling. And then it. Yeah, it also makes me more anxious, more and more anxiety, too. So, like that, that can contribute to like my mental stability, too. Right.”*
(C03: 23; Male; Housed; Non-Prescribed Substance Use: N/A; Prescribed Medication: N/A)

Although most participants did not mention experiencing or being diagnosed with a heat-related illness, one participant recalled a heat exhaustion diagnosis; however, they were unsure if it was solely or in part associated with drug use:


*“I remember this one time during the summer I was at the hospital, and I was so drained I’m so high out of my mind, and heat exhaustion was suggested.”*
(A14: 33; Male; Unhoused; Non-Prescribed Substance Use: Heroin, Cocaine, Crystal Meth)

#### 3.2.2. Impacts on Cognitive Capacity, Emotional and Mental Stability

It was also common for participants to share that they “got lazy in the heat” (A13: 24; Male; Housed; Non-Prescribed Substance Use: Meth, Fentanyl, Benzos) or experienced other sensations of lethargy or sleepiness. However, despite the descriptions of sleepiness and fatigue, many participants described experiencing a detrimental effect of heat exposure on sleep quality and duration. For example, participants shared the following experiences:


*“Yes, yes, yes. And like this, it has been verified by many studies already that sleep does have an effect...less quality sleep equals more chance of psycho-psychotic events happening in any mentally disturbed person. Actually, even in normal people, even normal people, if they get sleep deprived, if they get sleep deprived as like, they might even like have somehow like long enough to get like almost psychotic symptoms, like hallucinations and stuff right, but, this is more for the people who are mentally ill like who are already mentally ill or and then we need medication. Right.”*
(C03: 23; Male; Housed; Non-Prescribed Substance Use: N/A)

Participants described feeling more irritable, worried, panicked, or in general having negative feelings. However, the most common among the emotional impacts of the heat, as described by participants, was feeling anxious or experiencing frequent anxiety attacks. For example, participants expressed their experience with the following:


*“Anxious. Anxious. I feel really anxious when I’m hot. I can’t breathe. I can’t. Well, of course, I can breathe, but I feel like I can’t breathe. And I kind of feel very anxious and irritated. I’m worried. Um, like, negative feelings. And some people say, Oh, my God [heat] is the best thing ever. It’s finally hot and warm. It’s like, no, not for me.”*
(C05: 57; Female; Housed; Non-Prescribed Substance Use: N/A)

Participants also discussed the emotional toll that came with learning of the significant death toll from the 2021 Heat Dome, especially among people in shared networks (e.g., individuals from the same mental health clubhouse—local, community-based locations that offer people living with mental illness opportunities for friendship, employment, housing, wellness, etc.). Participants also shared that some of the psychological and emotional impacts associated with heat can make it challenging to stay on top of their schizophrenia treatments.

Lastly, although a less common experience, some participants described that the heat made them feel other emotions, like anger and annoyance, and heightened feelings of irritability, aggression, hostility, and desire to fight others. These cognitive effects were not uniform; the intensity of cognitive disturbances varied depending on the participant’s level of mental and emotional stress at the time. This variability underscores the need to examine how different symptoms, such as paranoia and psychosis, might interact with the cognitive effects of extreme heat.

#### 3.2.3. Worsening of Symptoms Associated with Schizophrenia

Many participants also shared experiences related to the worsening of schizophrenia-associated symptoms. Common among these symptoms was describing heat as a trigger for paranoia, psychosis, mental disturbances, delusional thinking and both auditory and visual hallucinations. For example, one participant linked their paranoia with the heat:


*“I think with schizophrenia, and with like, the paranoia that can come with it. I think if you’re concerned about something and it’s not going away, then that kind of ruminates, and you can get more worried about it. And if you have any other health concerns on top that would add to that and exacerbate it. And I think yeah, there’s many things you can worry about.”*
(C08: 40; Male; Housed; Non-Prescribed Substance Use: N/A)

A couple of participants attributed (at least in part) the relationship between the heat and worsening symptoms to their hydration level. For example:


*“I know it does, but not been able to like, quantify or qualify exactly…When I’ve been hot and overhyped, over dehydrated, and I’m unwell, or I’m not eating well, I noticed the experience the psychosis, definitely amidst the schizophrenia symptoms do appear.”*
(C04: 45; Male; Housed; Non-Prescribed Substance Use: Methamphetamines)


*“It would just like, the sweating and the malnutrition of it, and not having water. And I guess it just, seized my brain. You know, like, no water with the brain, you know.”*
(A01: age not specified; Male; Unhoused; Non-Prescribed Substance Use: Heroin, Cocaine, Speed)

Similar to the connection to hydration, other participants described that the general stress caused by enduring an EHE increases the likelihood of them experiencing acute symptoms. For example:


*“It wasn’t the setting of the activities that it was—it wasn’t like what we did that was the problem. It’s like the stress that stress-induced like heat inducing, or the heat caused me to get stressed out basically and then like lose my mind—and get out of touch with reality and start hearing, you know, the Gods in my head again…there were Gods in my head or like divine or entities trying to command me to do the good thing…but no, no, no, it’s just psychosis in my mind. Just psychosis, right? Voices and sometimes visions…It just increases the chance for those [with] schizoaffective, you know, makes it unlike normal people where they wouldn’t be hallucinating. Makes me more prone to these hallucinations. Because of my disorder, you know, because, especially during a heatwave, right, especially when the heat causes me stress, right.”*
(C03: 23; Male; Housed; Non-Prescribed Substance Use: N/A)

Some participants described that the worsening of symptoms was temporary and, once they returned to a cool environment, it dissipated, for example:


*“But yeah, I noticed my schizoaffective [disorder] is more, I hear more voices when I stay in the heat and don’t seek shade and don’t drink water. But I notice it more when it’s really hot out…It’s true that heat does affect schizoaffective disorder. We hear more voices. And it seems to bring it out a little bit more…puts you in kinda like a trance…you just kind of get stuck in the loop with the voices because you get so hot, you’re just too hot. But if you’re not in the blaring sun, that’s why I’m saying seek shade, because it’s when the sun is shining right on you, then you hear the voices. If you seek shade, then I think you’re fine. If you’re just hot, it’s fine, but if you got sun blaring at you, then you start to hear the voices. I think that has a lot to do with it.”*
(A09: 29; Female; Housed; Non-Prescribed Substance Use: Methamphetamine, Ketamine, Acid, NAS, Mushrooms)

In contrast, other participants described that they were able to manage their symptoms when exposed to the heat but, once they returned indoors, their symptoms were elevated, for example:


*“And when I was outside in the heat, I’d be too tired to have my schizophrenia go off. But when I was back inside, it would be going crazy as soon as the heat went away.”*
(A13: 24; Male; Housed; Non-Prescribed Substance Use: Methamphetamine, Cigarettes, Fentanyl, Benzos; Prescribed Medication: N/A)

Lastly, one participant also described that he self-selected to discontinue his prescription medications because “[his] schizophrenia was pretty bad”, denoting a counter-intuitive action that may lead to an exacerbation of schizophrenia symptoms.

#### 3.2.4. Impacts on Everyday Life

Participants also discussed how the heat caused disruptions to their daily routines. Concerning work, one participant shared that, because of the heat, they were unable to work because public transit did not have air conditioning, and other participants shared that they became physically ill at work due to the heat, as illustrated in the following quote:


*“And a few times got really lightheaded at work. Like I would get down to look in the drawers for a cosmetic or a cream or something and stand back up, I’m five nine and a half, I’d stand back up, and I feel dizzy and have to clutch onto one of the rails at work, but so I don’t know if that was dehydration or the stuff in the air.”*
(C06: 60; Female; Housed; Non-Prescribed Substance Use: Alcohol)

The heat was also described to have impacted participants’ ability to engage in hobbies and activities of daily living. Other participants had to modify their exercise regimes to avoid the heat and limit exposure to fitness facilities without air conditioning. As another example, one participant shared how the heat affected their capacity to cook and eat nutritiously:


*“I couldn’t cook. I remember that. I could not cook. Because if you turned the elements on or the stove on at all, it’s way too hot. We had to order Domino’s like every day.”*
(A10: 31; Male; Housed; Non-Prescribed Substance Use: Cannabis, Alcohol, Cocaine)

Lastly, a few participants attending college and university at the time of the 2021 Heat Dome experienced impacts from the cancellation of classes and their capacity to perform academically. For example, one participant shared:


*“And I was like, forgetting how to even though I’ve written tons of essays, I was like, forgetting how to... [write]. And it was, and I didn’t do everything properly….but I forgot how to construct an essay and, um, I guess, there was just a lot of fog, and I’m just, I’m not back to where I am not back to the ability I had before the heat dome.”*
(C02: age not specified; Female; Housed; Non-Prescribed Substance Use: N/A)

### 3.3. Theme II: Lack of Information

Some individuals expressed a personal lack of knowledge on managing extreme heat and employing adaptive strategies. For example, some participants shared their struggles more generally, expressing that they believed there was nothing they could do about the heat. In contrast, others gave more specific examples of the challenges they faced. For example, one individual shared that they were unable to use their mechanical cooling:


*“I couldn’t really figure out what was what, there was no manual [on the air conditioner] or nothing.”*
(A10: 31; Male; Housed; Non-Prescribed Substance Use: Cannabis, Alcohol, Cocaine)

A few participants also shared feedback that they *‘don’t know what they don’t know’* about how to manage heat, which also comes down to a lack of knowledge of heat mitigation strategies. For example, one participant said:

*“It’s hard for me to imagine what I need to be cooler*.”(C02: age not specified; Female; Housed; Non-Prescribed Substance Use: N/A)

Some participants felt there was a lack of information and awareness on the EHE, which further put their health at risk. Participants expressed that they did not know what community resources or services are offered to support people exposed to the heat (e.g., not knowing what cooling centres are or where they are in the community), where to access helpful information, or how warnings are shared with the public. Participants also shared that they had expected specific communication channels or networks to have provided them with health information. For example, one participant who was a student at the time said:


*“I’m kind of confused about why I didn’t see anything any warning there. I was just really upset that the College didn’t provide information on our news feed.”*
(C02: age not specified; Female; Housed; Non-Prescribed Substance Use: N/A)

This gap in knowledge and communication hindered participants’ ability to manage the heat exposure effectively and, for some, it resulted in no actions being taken to mitigate the heat. Instead, their approach was to maintain their routine and endure the heat. For example, when asked if they used any cooling strategies, one participant responded:


*“I didn’t. I braved it most of the time, like about 90% of the time, and I do basically anything anybody asked me to do. So there wasn’t really activities I was avoiding. But it wasn’t like, it wasn’t like, I was not happy to do it. I’m not gonna lie.”*
(A10: 31; Male; Housed; Non-Prescribed Substance Use: Cannabis, Alcohol, Cocaine)

Other similar sentiments were shared to express that, regardless of the extreme heat, activities and daily life had to continue for many participants.

### 3.4. Theme III: Cooling Strategies Employed to Mitigate the Heat

To help mitigate the potential for adverse outcomes from the heat, interviewees described employing various individual-level strategies to cool themselves (e.g., self-dousing with water), building-level strategies to cool their living space (e.g., opening windows) and other techniques suitable for alternative living environments (e.g., seeking shade). These strategies are further described below.

#### 3.4.1. Individual-Level Cooling Strategies

Participants most commonly described using individual-level strategies, such as taking more frequent, cool showers and consuming cool or cold drinks (e.g., water, pop) and foods (e.g., *“I heard about a radio station that [the heat dome] was coming…so I went and bought ice cream*” (Participant C07: age unspecified; female; housed; Non-Prescribed Substance Use: None), as key behaviours to reduce the impact of the heat. For example, one participant shared:


*“My husband would put me in the shower and run cold water over me. And so that was he would have done that three times a day.”*
(Participant C02: age not specified; female; housed; Non-Prescribed Substance Use: None)

The use of self-dousing, wetting clothing, cold towelling, and using ice packs were also frequently mentioned cooling strategies, for example:


*“And I ended up surviving just with theone powered fan that I have. But I also learned about new tricks. Like you can sleep with your cold socks or wet socks, and that will help regulate your body... It’s like, you know, like wet clothes or damp clothes. kind of help regulate your body.”*
(Participant C04: 45; male; housed; Non-Prescribed Substance Use: meth)

Some participants also described wearing less clothing or more light-weight and light-coloured garments, sweatbands, and cooling bandanas. Another common approach to staying cool was reducing physical activity where possible (e.g., taking public transit instead of walking) and altering daily schedules to minimize exposure to the sun and high-heat periods during the day in favour of cooler nighttime temperatures. Although less common, a few participants also mentioned accessing splash pads, getting their hair cut, limiting heat-generating appliance use, among others. For example one particpant shared the following strategy:


*“I do remember having to go and have a cold shower and, in my apartment lay down on the floor. Because the heat rises, right?... I was able to manage it by getting down closer to the floor and regulating it with fans.”*
(Participant C06: 60; female; housed; Non-Prescribed Substance Use: alcohol)

However, some participants reported increased difficulty using these strategies effectively, as their heightened schizophrenia symptoms made them reluctant to rely on public spaces or communal resources. This variability in coping effectiveness highlights the need for personalized interventions that consider the unique challenges faced by individuals with schizophrenia.

#### 3.4.2. Building-Level Cooling Strategies

For building-level strategies, some participants indicated they had access to methods of mechanical cooling, such as air conditioning in their homes, and activated it during the EHE. Some indicated they modified airflow in their living space using natural ventilation (e.g., opening and closing windows/doors based on indoor and outdoor temperature). For example, one participant shared:


*“Yeah, actually, I do remember; what we did was we would leave the windows open at night and shut them in the day, including the balcony door; we’d have that open at night and shut it in a day. My only concern with that was like if it was a security risk, but we just felt like the risk was outweighed by the benefit of having a cooler house.”*
(Participant C08: 40; male; housed; Non-Prescribed Substance Use: None)

The use of window shading, including blinds or other means to block sunlight, was also commonly reported. Lastly, although less frequently mentioned, some participants also attempted to wet the exterior surfaces of their home (e.g., the balcony) to help achieve a cooling effect (e.g., “During the heat, was pour water on the balcony to see if it would cool it” [Participant C02: age not specified; female; housed; Non-Prescribed Substance Use: None]).

#### 3.4.3. Cooling Strategies Used in Alternative Living Environments

Some participants shared that they were simultaneously experiencing homelessness or housing insecurity during the 2021 Heat Dome, previous EHEs, or hot weather periods. As a result, they relied on cooling strategies, such as seeking shade during the day, locating cool spaces to sleep outdoors (e.g., under tree canopies and bridges). For example, one participant shared:


*“I just tried to find a shady tree or something. I lived in the, in the parks underneath trees and stuff. So you know, I would just stay in the shade.”*
(Participant A02: age not specified; female; housed; Non-Prescribed Substance Use: Heroin, crytal meth, fentanyl)

Participants experiencing homelessness also described accessing cool public spaces for reprieve from the heat, such as community centers, grocery stores, restaurants, libraries, mental health clubhouses, malls, legions, gyms, and hospitals. Lastly, a few participants also mentioned that they used drugs in order to help mitigate the impact of the heat. For example, one participant said:

*“I ate cold foods. And sometimes I’d grab ice and put it on my head if I needed it. But otherwise, I just used drugs, and the heat just kind of didn’t seem as bad if I was high*.”(Participant C04: 45; male; housed; Non-Prescribed Substance Use: meth)

### 3.5. Theme IV: Participant Perspectives and Suggestions for Future Heat Protection

The participants also voiced many suggestions and ideas to enhance heat-health protection in the future. The strategies included individual behaviour-oriented mitigations and adaptation opportunities, like increasing access to resources (e.g., light-weight summer clothing) and education on heat-related illnesses and cooling strategies. The participants also shared ideas related to the built environment, including increasing access to green space and shade, as well as access to materials to improve at-home cooling (e.g., window film to reflect sunlight). Lastly, various suggestions were proposed for structural and societal adaptation opportunities, including improved training for healthcare providers, community cooling centres, and services for at-home check-ins during heat events, among others.

Some participants also mentioned they were *“surprised anyone cared*” (C02: age not specified; Female; Housed; Non-Prescribed Substance Use: none;) about people with schizophrenia and shared that they have often felt excluded by decision-makers on programming and service design that directly impacts them. One quote succinctly expresses this sentiment: “Why are we not listening to the people that have gone through it? They have so much to share” (C05: 57; Female; Housed; Non-Prescribed Substance Use: None).

## 4. Discussion

Through engaging directly with people with schizophrenia, we explored their lived experience with, and responses to, navigating an EHE—the 2021 Heat Dome. Although, in some ways, the experiences described by our participants concerning the impact of the EHE were similar to the experiences reported among other heat-vulnerable populations and the general public, our data also showed that the participants’ experiences were uniquely socially patterned and underpinned by the presentation and manifestation of their schizophrenia.

The biopsychosocial model allows a more holistic understanding of the experiences of heat by individuals with schizophrenia. Biologically, extreme heat exacerbated physical manifestations and contributed to a worsening of schizophrenia symptoms, such as increased paranoia, hallucinations, and cognitive disturbances. Participants reported heightened psychotic episodes and difficulty managing their symptoms in response to heat exposure, illustrating how physiological stressors can interact with underlying mental health conditions. Psychologically, the heat led to increased anxiety, irritability, and disrupted sleep patterns, which further compromised mental stability and daily functioning. These emotional responses, coupled with cognitive fatigue, underscore the psychological burden imposed by environmental stressors. Socially, the impact of heat on participants was influenced by factors such as housing status and access to resources. Those experiencing homelessness faced greater challenges in finding cooling solutions and maintaining health, reflecting the significant role of social determinants in shaping heat-related health outcomes. In addition, participants received varying levels of community support, primarily through intensive case management or assertive community treatment teams. Both approaches involve inter-disciplinary team-based services, including access to psychiatrists, social workers, nurses, and case managers, additionally providing limited outreach. The level of interaction between patients and service providers varied substantially. Moreover, the lack of information and resources, combined with feelings of exclusion from decision-making processes, highlights the importance of addressing social and systemic barriers to improve resilience and support for vulnerable populations during EHEs.

Overall, this model emphasizes the interplay of biological, psychological, and social factors in understanding the multifaceted impacts of extreme heat on individuals with schizophrenia and underscores the need for multi- and inter-disciplinary approaches in healthcare. This highlights that effective treatment and support require consideration of the whole person rather than isolated symptoms.

Below, we reflect on these findings and propose how public health strategies, policies and future research could consider the contexts, situations, needs and challenges that may increase protections for people with schizophrenia during EHEs.

### 4.1. The Presentation and Expression of Heat Stress by People with Schizophrenia

The participants’ descriptions of the physical impacts of heat stress were similar to those conveyed in public health guidance for the general population, including signs and symptoms like excessive perspiring, light-headedness, dizziness, and fatigue. However, a unique characteristic of our participants’ experiences was the worsening of symptoms common to schizophrenia (e.g., heightened paranoia, more frequent hallucinations), as well as how they described or expressed these symptoms. For example, some described heat stress as ‘you are falling out of one’s body’ or ‘feeling outside of reality.’ This illuminates an important opportunity to draw awareness that people with schizophrenia experiencing adverse effects of heat may not be expressing verbally the common cue words others may expect and associate with the continuum of heat-related illness. In other health disciplines unrelated to heat, authors have suggested that individuals with schizophrenia often have a reduced capacity to communicate important medical symptoms due to a combination of cognitive deficits, disorganized thinking [40], and a decreased self-awareness of changes to their physical health [41]. Further, individuals with schizophrenia often face challenges with communicating clearly (e.g., narrative, intonation, and gestures) [42] and present with ‘theory of mind’ deficits, which are related to a lack of expressiveness [43,44]. Therefore, how our participants expressed and described their experiences with heat stress may point to the need to further investigate how individuals with schizophrenia communicate or describe the early signs of heat-related illness. This type of investigation could lend to the identification and need to develop different screening options to prevent progression to severe or fatal outcomes.

### 4.2. Implications of Co-Occurring Heat Stress and Schizophrenia Symptoms

Participants also shared that some of the symptoms they experienced are related to their schizophrenia—separate from the heat. For example, altered cognitive functioning and behaviour change is a key sign of the onset of a heat-related illness and its progression to a more severe condition [42]; it is also well known that cognitive deficits form a major part of the schizophrenic syndrome [45]. Similarly, some participants recounted feeling agitated and combative during the 2021 Heat Dome, comparable behavioural reactions that are congruent with experiencing heat stress [46,47,48]. However, such behaviour aligns with stereotypical and misinformed portrayals of people with schizophrenia in media and popular culture [45]. Consequently, healthcare providers, family members or the public may misattribute these behavioural changes to one’s diagnosis rather than recognizing heat as the cause. This misunderstanding could potentially lead to the progression of a heat-related illness to a more severe or fatal outcome (i.e., heat stroke) if left untreated [46]. Additionally, it is also well established that schizophrenia is a heterogeneous clinical syndrome, and the presentation or manifestation of this disorder can vary extensively between individuals [47]. This emphasizes the need for education on identifying the unique presentation of heat stress, the individualized nature of monitoring and prevention initiatives, and improved diagnostic tools that can differentiate between these overlapping experiences, all the while recognizing the range of individual variability in how people respond to heat in any given group. Furthermore, there are many reasons for variability, including a self-guided decision to stop their medication (as one study participant had); accounting for these factors and finding ways to prevent them will be crucial in the path forward.

### 4.3. Navigating the Intersection of Substance Use, Schizophrenia and Heat Stress

Many of our participants also shared that they regularly engaged in substance use at the time of the 2021 Heat Dome and wove together stories of being under the influence of drugs or alcohol while experiencing heat stress. The intersection of psychosis, substance use, and heat stress symptoms presents a complex challenge. Distinguishing between symptoms of schizophrenia, substance use effects, and heat stress can also make it difficult for both the individual and healthcare providers (e.g., first responders) to identify and respond appropriately. The challenge lies not just in identifying the symptoms associated with schizophrenia or the effects of substance use, but also in recognizing the progression from mild to severe heat stress. This task is made even more complicated when individuals with schizophrenia who use substances lack the awareness or capacity to differentiate between these varying levels of heat stress, potentially overlooking the signs of heat-related illnesses until they reach a critical point. Again, this emphasizes the need for future research and initiatives to communicate the compounding risk of substance use during extreme heat. However, it is important to acknowledge that simply adopting existing messaging on substance use in the heat developed for the public may not be adequate for this population, as it ignores additional layers of complexity that must be considered. For example, Polgar et al. [49] found that people with schizophrenia did not appear to be receiving benefits from generalized smoking-related public health measures. The authors proposed that one factor that could be impacting the effectiveness of the public health measures for this population was the purpose of smoking. They found that, for some people with schizophrenia, smoking was not simply a vice or a habit but a means of obtaining the medication necessary for symptomatic relief of their disorder [49]. As seen by our participants, substance use was, at times, used as a heat-mitigation strategy or a coping strategy to deal with the heat event. Thus, public health measures should consider the evidence concerning the specific needs and social vulnerability circumstances of people with schizophrenia.

### 4.4. Implications of Information Gaps on the Heat Event and Cooling Strategies

Many study participants also indicated a lack of awareness of the heat event before it started, or even during the peak of the heat, and reported that they did not employ any cooling strategies. In parallel, some of our participants indicated that, due to fear and distrust of government agencies, they ignore the news—a primary communication channel for disseminating heat warnings and protective measures in Canada [50]. At the same time, another challenge arose for some individuals for whom knowing about the impending EHE triggered or exacerbated their anxiety. Taken together, locating the appropriate and trusted medium for communication while avoiding triggering paranoia, anxiety, or distress poses a unique challenge for providing timely and effective EHE information to individuals with schizophrenia [42]. Translating these findings to extreme heat protection may indicate that our current heat alert response system communication channels and other effective heat interventions in the general population may not be effective for people with schizophrenia [51]. This points to the importance of establishing or leveraging existing trusted communication channels and relationships to facilitate the dissemination of heat warnings and evidence-based heat protective strategies. For example, providing educational resources to areas that have high schizophrenic populations, such as low-income housing, emergency healthcare units, and areas with a greater density of people experiencing homelessness, may be beneficial, as it increases the awareness and resources available to those in close contact with schizophrenic populations. At a policy level, increasing communication and educational tools about EHEs for people with schizophrenia can promote the trust and rapport necessary for information to be disseminated [42]. This would ensure that messaging is reaching this population and is increasing awareness and, where applicable, initiating behaviour change for future heat events. Further work will be needed to assess novel ways of presenting information using alternative and directed strategies.

### 4.5. Addressing Discrimination and Stigma in Health Care for People with Schizophrenia and Its Implications for Heat Protection

Our participants also shared that, when managing in the heat, they did not want to ask for help, expressed tendencies to avoid medical care or other public resources like cooling centres, and shared stories of experiencing discrimination. This is consistent with research from different health disciplines (e.g., cancer care, dental health) [52,53], which has demonstrated that people with schizophrenia may anticipate that healthcare providers will discriminate against them [54] and are therefore less likely to seek medical care or disclose their physical symptoms [41]. These perceived barriers may decrease the access to and delivery of care for individuals with mental disorders [40,55]. Although there is little evidence specific to heat, previous studies have shown that some of these real and/or perceived barriers present as medical professionals’ being more likely to attribute medical symptoms to the psychiatric illness (as opposed to potentially another, separate health issue such as heat stress) and subsequently being less likely to refer individuals for further care or support services [56]. This is critical in extreme heat protection because, if individuals do not feel comfortable presenting to the hospital due to actual or perceived stigma and discrimination (i.e., receive substandard treatment), the likelihood of worsened heat-related health outcomes increases. Similarly, other studies have shown that, in the families of patients with schizophrenia, internalized stigma can also discourage help-seeking and may also result in families who attempt to provide care themselves without assistance from mental health services [57]. As such, enhancing messaging to the broader population on heat vulnerability factors that may exacerbate risk for people with schizophrenia (e.g., medication use, clothing layering, lower awareness of physical manifestations), as well as addressing the stigma surrounding schizophrenia overall, may enhance heat-health protection for those living with schizophrenia. Although yet to be explicitly assessed for schizophrenia and heat, other health disciplines have found similar strategies of promoting prevention by reinforcing the adverse impacts on particular populations to the general public influential in broadening the reach of messaging [54].

### 4.6. Participant-Driven Suggestions for Future Heat Protection

Participants offered several practical suggestions for enhancing heat protection for individuals with schizophrenia. They emphasized the need for improved access to resources, such as light-weight summer clothing and cooling materials like window film, to manage heat exposure more effectively. Additionally, targeted education on heat-related illnesses and effective cooling strategies, along with enhanced training for healthcare providers to recognize and address heat stress in this population, were deemed crucial. Participants also highlighted the importance of increasing access to green spaces, shaded areas, and community cooling centers, as well as establishing at-home check-in services during heat events to offer essential support. Furthermore, they stressed the value of involving individuals with schizophrenia in the planning and design of heat protection programs to ensure that their specific needs are addressed and to foster trust in the system. Integrating these participant-driven suggestions into public health strategies could lead to more effective and inclusive measures, thereby improving the resilience and well-being of individuals with schizophrenia during EHEs.

The heat protection practices offered by participants provide four areas for consideration:Access to Resources: Ensuring availability of light-weight summer clothing and cooling materials, such as window film, to help manage heat exposure.Education and Training: Providing targeted education on heat-related illnesses and effective cooling strategies and improving training for healthcare providers on recognizing and addressing heat stress in people with schizophrenia.Community Support: Enhancing access to green spaces, shaded areas, and community cooling centers. Establishing services for at-home check-ins during heat events can also provide crucial support.Inclusive Policy Development: Actively involving individuals with schizophrenia in the planning and design of heat protection programs to ensure their needs are adequately addressed and to build trust in the system.

These findings emphasize the essential need to bring people with schizophrenia to the forefront of the redesign and development of public heat response initiatives and implementation of protective measures. This includes developing tailored individual- and community-level heat response along with communication initiatives and strategies to address the layers of vulnerability. However, as mentioned above, it has been shown that broad public health programming, policies and interventions that appear to be effective in changing behaviour in the general population may not translate to people with schizophrenia [51,58,59]. Drawing from the broader public health literature, other strategies have been used to design public health initiatives for people with schizophrenia that could be considered for heat mitigation programming. For example, to address the high rates of dental concerns among people with schizophrenia, a group in Australia developed a collaborative outreach dental program where teams take mental and oral health services to ‘hard-to-reach’ settings such as rooming houses and supported housing to eliminate the need to report to a clinic [52,53]. Considering such strategies for sharing heat health information with people with schizophrenia living in community settings could not only address the concerns regarding media avoidance, but also address the unique living circumstances that many of the participants shared.

### 4.7. Limitations and Future Research Directions

As this study engaged with individuals with schizophrenia who survived the 2021 Heat Dome, our participants do not represent all individuals with schizophrenia who lived in B.C. during the 2021 Heat Dome, particularly those who may have experienced adverse morbidity or mortality outcomes due to the heat. Among the survivors there may also be compromises in recall, a common symptom of schizophrenia. This may exacerbate the existing challenges in recalling an event that occurred nearly three years prior. We further recognize that there may have been adverse emotional effects of discussing potentially traumatizing events, and informing people with schizophrenia of their additional vulnerability during EHEs could have a mental toll on participants. This is particularly crucial to recognize with a population who are starkly aware of their vulnerability. Further, there can be unintended consequences of data sharing, such as compromising the anonymity of participants from them sharing specific events or living situations. While our recruitment focused on those not in supervised healthcare facilities during the 2021 Heat Dome, many of our participants were also in various housing or other facilities with support services, which fortunately helped protect them. Therefore, although these findings contribute to developing a better understanding of why individuals with schizophrenia are at greater risk of death during EHEs than the general population in B.C., our sample is an incomplete representation of all individuals with schizophrenia in B.C. during the 2021 Heat Dome.

This study did not focus on the impact of medication on heat stress responses, which remains an important area for future research. While medications, particularly antipsychotics, are known to influence heat tolerance—potentially through dopaminergic, sedating, or anticholinergic effects—the specific variations in impact are not yet fully understood. Antipsychotics, such as haloperidol and clozapine, may affect thermoregulation differently, and the variability in response due to individual differences and polypharmacy adds complexity. Additionally, adherence to medication can vary among patients, influencing the results. Future research should address these factors to provide a more comprehensive understanding of medication effects on heat stress.

Lastly, as this study was a qualitative investigation based on a small sample of interviews, it was not within the scope of this work to present quantitative data for the purposes of more generalizable inferences (e.g., most frequently cited cooling strategies). Thus, we propose that a future survey-based investigation may be able to extend these findings further, as there is still a significant need to conduct further inquiries to address the remaining gaps in our collective knowledge.

## 5. Conclusions

This study explored the experiences of people with schizophrenia during the 2021 Heat Dome in B.C. provides critical insight into the complex ways that individuals with schizophrenia experience and respond to EHEs and reveals various situational and contextual factors that can further compound the challenge for some. Participants shared critical insight on how EHEs impacted them, including descriptions of mild to severe physical manifestations of heat stress (e.g., fainting, heat rashes), the triggering of schizophrenia-related symptoms (e.g., paranoia, hallucinations), and the determinantal effects on their energy levels and emotional stability, which further caused disruptions to their everyday life (e.g., work, activities of daily living). Participants also illustrated gaps in knowledge and challenges experienced with accessing information, which hindered their ability to manage the heat exposure effectively and, for some, resulted in no actions (or counter-intuitive actions) being taken to mitigate the heat. These results emphasize the need to create, or at least tailor, interventions, content, and mechanisms of communication to meet the different needs and beliefs of people with schizophrenia. This calls for greater collaboration between researchers, public health practitioners, policymakers, medical teams, other community stakeholders and, most importantly, people with schizophrenia themselves to improve heat-health outcomes.

## Figures and Tables

**Table 1 ijerph-21-01108-t001:** Visualization of the coding structure.

Themes	Personal Reflections on the Impact of Extreme Heat				Lack of Information	Cooling Strategies			Proposed Strategies for the Future		
Concepts	Physical Impacts of Heat	Impacts on Cognition and Mental Stability	Worsening of Schizophrenia Symptoms	Impacts on Everyday Life	Lack of Information and Awareness	Individual-Level Cooling Strategies	Building-Level Cooling Strategies	Cooling Strategies Used in Alternative Living Environments	Individual Behaviour Oriented Mitigations and Adaptations	Structural and Societal Mitigations and Adaptations	Built Environment Oriented Mitigations and Adaptations
Sub-Groups	- Challenges with Breathing - Sweating - Heat-Related Illness	- Adhering to Treatment - Emotional Toll - Aggression and Violence - Stress - Sleep - Mental Capacity - Lethargy - Disorientation - Anxiety	- Trigger for Psychosis - Discontinued Medication Use - Trigger for Hallucinations - Dehydration	- Work - School - Hobbies - Other	- Lack of Awareness - Lack of Knowledge of Strategies	- Electric Fans - Reducing Time Outdoors - Lying on the Floor - Self-Dousing - Limiting Heat-Generating Appliance Use - Reducing Activity Level - Consuming Cool Drinks and Food - Optimizing Clothing - Water Immersion - Hair Cut	- Mechanical Cooling - Window Shading - Natural Ventilation - Wetting Surfaces	- Seek Shaded Areas - Access Cool Public Spaces - Locate Cool Sleeping Areas - Using Drugs	- Education - Personal Resources	- Public Awareness of Schizophrenia - Representation on Committees - Home Cooling Evaluation Services - Peer Support and Social Interaction - Outreach Services and Foot Patrol - Enhanced Alerting - Home Check-In Services - Community Cooling Resources - Training for Healthcare Providers	- Home Modification - Green Space

## Data Availability

Data are available from the corresponding author (Liv Yoon, liv.yoon@ubc.ca) upon reasonable request and a signed access agreement.

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
