# Peer review of "Responding to the Heat and Planning for the Future: An Interview-Based Inquiry of People with Schizophrenia Who Experienced the 2021 Heat Dome in Canada"

_ijerph, 2024, doi:10.3390/ijerph21081108_

Round 1

Reviewer 1 Report

Comments and Suggestions for Authors

Author Response

We sincerely thank the Reviewing Editor and expert Reviewers for making a detailed critical assessment of our manuscript. The comments provided have led to significant improvements in our communication. Please see attached our responses. In all instances, reviewer comments are provided in 11 pt. Calibri font and the author responses directly following using 11 pt. Calibri font in bold. New and or revised text is presented in red italicized font. Please note that a marked and a clean version of the manuscript has been provided. 

Reviewer 2 Report

Comments and Suggestions for Authors

Dear Authors,

This manuscript presents a very pertinent and original theme and is extraordinarily well presented and written. In my review I did not detect changes when carried out. I suggest publication without changes.

Additional comments attached.

Many congratulations on your work and success for the publication.

Sincerely

Author Response

(The authors gave the same response as above.)

Reviewer 3 Report

Comments and Suggestions for Authors

Dear Author(s),

The manuscript entitled ‘‘Responding to the Heat and Planning for the Future: An Interview-Based Inquiry of People with Schizophrenia who Experienced the 2021 Heat Dome in Canada’’. This research, which was conducted in individuals with schizophrenia who experienced the Heat Dome in Canada, will contribute to the field in terms of including recommendations for the future.  Essentially,  It is a well-written article conducted in accordance with qualitative research criteria, which I enjoyed reading. I congratulate you all on the exciting manuscript. However, there are a few important points that need to be clarified. Please find my comments and feedback below with reference to lines:

1. Introduction

There is a well-written introduction that explains the importance of the research topic and the gaps in the field. I only suggest that authors pay attention to the following point. Please pay attention to this point throughout the full text if I have missed it.

Line 60- Reference 11 reviews the literature on symptoms following dosage reduction or abrupt discontinuation of neuroleptics. There is no explicit reference to schizophrenia. This reference should be removed from here. Reference 11 should be added at the end of the following sentence.

Line 60-61 ‘’These impairments have been largely attributed to the psychotropic medications typically prescribed to manage schizophrenia’’

2. Materials and Methods

Lines 100-105 The title ''2.1. Ethics'' can be included after the data analysis, and it should be given separately at the end of the article according to the journal's guidelines.

Lines 126-130 I suggest that they include the semi-structured interview guide mentioned here in the supplementary material, because each question in interviews with schizophrenic patients is important in terms of being non-threatening for these patients.  I appreciate that the authors were able to get so much information from the patients. It would be great to see the guide in the supplementary material so that this study can be adapted and replicated in other cultures.

Line 149 The data analysis section is theoretically well explained, but it would have been good to see more information on which themes the authors identified and which sub-themes they derived.

3. Results

The Results section and subheadings are written in accordance with the purpose of the research and reflect the essence of the subject well.

In Table 1, it could have been written how many participants gave these answers for cooling strategies. Because more than one participant can suggest a cooling strategy. And it would be good to know which methods are more often suggested by schizophrenia patients. For this reason, it would be more explanatory if it was given as follows, for example Using Electric Fans=3 and Wet clothing=7. The same frequency can be given for table 2 and table 3.

4. Discussion

The discussion section was written with its subheadings in line with the results of the research, with the authors making inferences and referring to the literature.

Author Response

(The authors gave the same response as above.)

Reviewer 4 Report

Comments and Suggestions for Authors

Review Report

Summary: The purpose of this article is to examine how people with schizophrenia experienced and were affected by the extreme heat events (EHEs) known as the 2021 "Heat Dome" in British Columbia (B.C.), Canada. Specifically, the study aims to understand the physical, cognitive, and emotional impacts of this event, as well as the level of awareness and protective health measures taken in response to EHEs.

The use of semi-structured interviews enabled a detailed understanding of individual experiences, providing a multifaceted view of the challenges. The findings of the study can support the development of initiatives and strategies to protect against heat, both at an individual and community level, specifically tailored to people with schizophrenia. However, the limitations identified must be recognized as significant barriers to the generalization and application of the findings. These criticisms emphasize the urgent need to address these limitations in future studies to better inform policies and practices aimed at effectively protecting people with schizophrenia during extreme heat events.

Scope: Due to the description of how people with schizophrenia respond to extreme heat events, this paper appears suitable for publication in the International Journal of Environmental Research and Public Health. It addresses a topic of public and environmental health importance, uses an appropriate methodology to investigate qualitative questions, and has the potential to make an important contribution to the existing literature on vulnerability in extreme weather conditions

Novelty: The central question of the study is clearly defined: "How do people with schizophrenia experience and respond to extreme heat events (EHE)?" Although the research question is not unique in every respect, it is original in that it focuses on a specific population group that is underrepresented in studies on EHEs. The combination of the specificity of the population studied the qualitative methodological approach and the potential contribution to knowledge on vulnerability to extreme heat makes the study question original and relevant.

Quality: The manuscript is clear, relevant to the field, and appropriately written and presented.

Level of English: The English language is appropriate and understandable.

Reader interest: Based on the aims and scope of the journal IJERPH, the results of the study on how people with schizophrenia respond to extreme heat events are potentially of interest to the journal's readers. Although the target audience is rather specific compared to more general studies, the study offers relevant contributions to the understanding of environmental influences on the mental and physical health of this population group and is thus in line with the journal's interests and mission.

Article: Because the study focused only on a specific community within B.C., Canada, the results may not reflect the experiences of people with schizophrenia in different climate environments or social contexts. Furthermore, the use of semi-structured interviews can introduce response bias, where participants may not accurately report their experiences or may be influenced by the wording of questions.

The study may not have considered all external factors that could influence participants' responses, such as access to financial resources, family or community support, and general mental and physical health status. Interviews were conducted between October 2023 and February 2024, more than two years after the 2021 extreme heat event. This time lag may have affected the accuracy of participants' memories of their experiences during the EHE.

While the focus on schizophrenia is important, the study did not compare results with other vulnerable populations, which could provide broader context about how different mental health conditions respond to extreme heat events.

SPECIFIC COMMENTS

Introduction: Although the literature review provides a comprehensive overview of issues related to the vulnerability of people with schizophrenia during EHEs, it lacks an explicit theoretical framework that could structure and guide data analysis and interpretation. A theoretical framework could help situate the study within a broader theoretical context, such as social vulnerability theories, stress response models, or health behavior theories, providing a conceptual framework to better understand the experiences and underlying mechanisms.

Although the study mentions schizophrenia and schizoaffective disorder, a clear definition of these terms in the context of the study would be helpful, especially for readers who are not experts in the field.

Some studies cited could be contextualized more explicitly in relation to the specific objective of the study. For example, how do these studies contribute to understanding individual experiences during EHEs?

While the importance of filling knowledge gaps is mentioned, a more detailed explanation of why these gaps are critical to public health and policy development would be beneficial.

Method:

Participants: Although the inclusion of a wide range of care settings (such as health centers and community facilities) enriched the representation of the sample, it would be important to clarify how this diversity of settings affected consistency in data collection and interpretation of results.

It is equally critical to clarify how eligibility criteria were applied consistently across different recruitment sites to ensure that the sample adequately represents the target population of people with schizophrenia.

To improve the understanding and applicability of the results, it would be recommended to also provide a more detailed and stratified description of the participants, including socioeconomic information, specific details about the medication and additional substances used, as well as a more critical analysis of the representativeness of the sample in relation to population of individuals with schizophrenia in similar contexts. This would help to better contextualize the results and their implications for different subgroups within the study population.

Finally, because many participants were in housing or other facilities with supportive services during the heat event, it is important to clarify how these conditions may have protected participants from certain heat-associated risks, and how results should be interpreted for people with schizophrenia who live in different housing or support conditions.

Ethical Issues: Obtaining ethical approval from multiple committees reinforces the study's commitment to rigorous ethical standards and compliance with local and national regulations. However, it is essential to provide detailed information about how ethical principles have been implemented in practice, including aspects such as informed consent, participant privacy, handling of sensitive data and protection against potential harm. Was it done differently depending on the data collection procedure: in person or virtually?

There is no specific mention about the disclosure of potential conflicts of interest between researchers, sponsors and institutions involved in the study, which could affect the credibility and objectivity of the study.

Procedure: Both face-to-face and virtual interviews (via Zoom or telephone) were conducted as part of the study, which may lead to differences in the quality and depth of interactions. Face-to-face interviews generally allow for richer interaction and make it easier to read facial expressions and body language, which can enrich the understanding of participants' responses. Virtual interviews can be limited by the quality of the connection, external interference and the difficulty of building as strong an empathic connection as face-to-face interviews. It is important to clarify the added value of using these variants of data collection methods. It must be clarified whether there were significant differences in the quality of the data obtained between face-to-face and virtual interviews. This includes assessing whether certain types of information were better explored in one interview format than in another. Are there differences in the way participants behave and express themselves in face-to-face and virtual interviews? Could this have affected the validity of the responses collected? There are also significant differences in the length of the interviews (15-120 minutes). Clarify whether there was a type of data collection where respondents tended to take less time to answer and discuss what this might mean and how it might affect the results.

Data analysis: While descriptive thematic analysis is useful for identifying patterns of meaning within a dataset, it has some limitations. The descriptive nature of thematic analysis can lead to greater subjectivity in the identification and interpretation of themes. Depending on the researcher's interpretation, the same data may lead to different themes, which affects the consistency and objectivity of the results.

The lack of an explicit theoretical framework also compromises methodological rigor. This can be particularly problematic when there are no clear criteria for selecting and defining emerging themes. Without a sound methodological framework, data coding may be less systematic, making it difficult for other researchers to replicate the study or fully understand the analysis process. The validity and reliability of the results may also be challenged by the lack of additional methods to triangulate or verify the identified themes. To address these criticisms, it would be advisable for researchers to provide a detailed and transparent description of the process of thematic analysis, including the criteria for inclusion and exclusion of themes, reflective discussions of the subjectivity involved, and considerations of the internal and external validity of the results obtained. In addition, complementary methods such as data triangulation or the involvement of multiple researchers in the analysis could be included to increase the robustness of the qualitative results.

Results: The results are organized into thematic sections, which is generally well suited to the presentation of qualitative data. However, a table showing the 6 themes covered in the interview and the different themes and sub-themes that emerged from each theme could be included, as well as clearer sub-headings or a better defined structure within each thematic section to improve readability and navigation. .

The inclusion of descriptive quotes is beneficial in anchoring the findings in the participants' voices, but the flow of the presentation could be improved with more coherent transitions between quotes and analysis. Although the thematic analysis captures a wide range of experiences related to the impact of extreme heat on people with schizophrenia, there is a need for deeper analysis within each theme. For example, further exploration of the ways in which certain symptoms (e.g. paranoia, psychosis) manifest in different contexts of heat exposure would lead to more comprehensive insights. The analysis could benefit from more nuanced interpretations of the links between different themes and subthemes. This would help to better understand the interaction between physical, emotional and cognitive effects.

Discussion: The approach to discussion in the document needs to be much more analytical. For example, many people with schizophrenia take medication. It would be important to investigate how different antipsychotic medications may affect the response to heat stress and whether there were differences in the symptoms of participants taking different types of medication.

Although the need for adapted mitigation strategies is mentioned, it would be interesting to discuss specific examples of how these strategies could be implemented in practice. For example, how heat warning systems can be adapted to be more accessible and effective for people with schizophrenia.

Considering that people with other severe mental disorders may also face similar challenges, it would be interesting to discuss how the experiences of people with schizophrenia during heat stress compare to these groups. This could improve understanding of the specific needs of different populations.

Although it is mentioned that educating the public about the specific needs of people with schizophrenia is crucial, it would be useful to discuss concrete strategies to raise public awareness and reduce stigma associated with schizophrenia, especially in extreme heat emergency care.

How the findings of this study could inform public policy and health practice, such as improving emergency services, health professional training and housing policy, would also be a valuable addition to the discussion.

Limitations: Lack of explicit discussion of potential effects of the study on participants, such as emotional distress or unintended consequences of data sharing.

References: Most of the sources cited are recent and relevant, particularly those discussing the impact of heatwaves on mental health and the vulnerability of certain populations, such as the homeless. However, there are also some older references that can still be very relevant to understanding certain aspects of schizophrenia and its impact on treatment.

Overall merit: The study focuses on a significant gap in research on how people with schizophrenia cope with extreme heat. It not only adds to the body of knowledge on the impact of extreme heat on mental health, but also provides a solid foundation for future interventions that can improve the safety and well-being of people with schizophrenia during extreme weather events. These global benefits underline the relevance of the study for science as well as for clinical practice and policy.

General recommendation: Reconsider after Major Revisions.

Comments on the Quality of English Language

The overall quality of the English in the article is satisfactory, with clear and comprehensible language throughout. However, there are some areas that could be improved to enhance clarity and readability:

 - There are instances where sentence structure could be simplified to improve readability. For example, long sentences with multiple clauses can be split into shorter sentences.

 - Ensure consistent use of tense throughout the article. There are occasional switches between past and present tense that could be harmonized.

 - Some terms and phrases could be made more precise. For example, instead of "impairments in their physiological ability to activate heat loss responses"," it could be "impairments in their ability to regulate body temperature"

 - Avoid jargon or overly complex vocabulary that may not be accessible to all readers. For example, using "EHEs" (extreme heat events) without prior definition may confuse some readers. Always define acronyms the first time you use them.

 - Some sections contain redundant wording that could be streamlined. For example, "Given that less than 1% of Canadians are diagnosed with schizophrenia [4], these numbers are extremely concerning" could be simplified to "Given that less than 1% of Canadians are diagnosed with schizophrenia, this is extremely concerning."

 - Make sure each paragraph conveys a single clear idea. Some paragraphs could be restructured to ensure a logical flow of information.

Author Response

(The authors gave the same response as above.)

Round 2

Reviewer 1 Report

Comments and Suggestions for Authors

2nd Review: “Responding to the Heat an Planning for the Future: An Interview-Based Inquiry of People with Schizophrenia who Experienced the 2021 Heat Dome in Canada”

The authors took great care to address all issues raised by my first review. In its present form, especially with the supplementary materials provided, I deem the article to be of good scientific quality and ready to be published after addressing a few minor corrections and suggestions. I would like to express my deep-felt respect for how the research team enabled a marginalized group to add their valuable perspective regarding a current and important topic such as societal adaption to future EHEs.

Figure 1: Although I admire the aesthetics of this figure, I do not think its format is suitable for presentation as part of an article, as its almost impossible to read the subcategories, even after zooming in. Please consider choosing a more easily accessible version to present your results, for example a simple table with the themes as the 3 sub-headings and two columns, one for concepts, one for sub-groups per concept.

Line 497: You could add an example of a substance you classified as an ‘additional substance’ to give the reader an idea of what this might entail without having to consult the supplementary material.

Results: During my first review, I didn’t notice that you specified the medication of each participant when quoting them. For one, you forgot to do this for the new quotes from section 3.4.1 onwards. However, I would question the necessity of this information. While it is quite informative to have an impression of the great variety of substances given in the supplementary material, adding this information to each quote suggests that the participants’ medication and substance use is of particular importance in understanding their experiences. In contradiction to the bio-psycho-social approach you’re describing, this would put a strong emphasis on the biological aspect of a life with the diagnosis of schizophrenia. Thus, I’d argue in favor of removing the information for all quotes in the result section.

Supplement A: My version of the file contains corrections, please remove these for the final version.

Supplement B: As all codes are descriptive and inductive, you could consider giving this information in the description of the table and deleting the respective columns.

Supplement C: Table 3 is a bit confusing and I would ask you to alter it, adhering to a medical perspective. Two aspects should be modified: first, you give “depo shot” and “sleeping pills” as medication, as well as two instances where you are not certain. I would summarize all of these cases as “unknown”, as different kinds of medication can be given out as a depot, and the same goes for sleeping pills. Secondly, please standardize the names and only give the name of the substance, not the name under which the medication is sold. For instance, Seroquel is the drug name for quetiapine, those two are the same. This makes the information much more accessible to an international readership, as drug names are often country-specific, e.g. Adderall.

Author Response

Please see attached our responses to Reviewer 1.

Reviewer 4 Report

Comments and Suggestions for Authors

I would like to express my appreciation to you for taking the trouble to include all my suggestions in the article. I believe that the changes have considerably improved the content and quality of the work.

I am confident that with these revisions, the article is now in excellent shape for publication.

Congratulations on this great work!

Author Response

Please see attached our responses to Reviewer 4.
